# MID-POSE: Multi-Instrument Detection and Pose Estimation in Endoscopic Surgery

**Wenhua Wei**[*1] (iD)                                                     WENHUA.WEI.17@ALUMNI.UCL.AC.UK

**Laurent Mennillo**[*1,2,3] (iD)                                                     L.MENNILLO@UCL.AC.UK

**Zhehua Mao**[1,2] (iD)                                                     Z.MAO@UCL.AC.UK

**Anjana Wijekoon**[1,2] (iD)                                                     A.WIJEKOON@UCL.AC.UK

**Kendall Feeny**[1,2] (iD)                                                     K.FEENY@UCL.AC.UK

**Danyal Zaman Khan**[1,2] (iD)                                                     D.KHAN@UCL.AC.UK

**Evangelos B. Mazomenos**[2,3] (iD)                                                     E.MAZOMENOS@UCL.AC.UK

**Danail Stoyanov**[1,2] (iD)                                                     DANAIL.STOYANOV@UCL.AC.UK

**Hani J. Marcus**[2,4] (iD)                                                     H.MARCUS@UCL.AC.UK

**Sophia Bano**[1,2] (iD)                                                     SOPHIA.BANO@UCL.AC.UK

[1] *Department of Computer Science, University College London, London, United Kingdom*

[2] *UCL Hawkes Institute, University College London, London, United Kingdom*

[3] *Department of Medical Physics & Biomedical Engineering, University College London, London, United Kingdom*

[4] *Department of Neurosurgery, National Hospital for Neurology and Neurosurgery, London, United Kingdom*

**Editors:** Accepted for publication at MIDL 2026

## Abstract

Reliable perception of surgical instruments is a key prerequisite for intraoperative guidance, context-aware assistance, and workflow analysis in minimally invasive surgery (MIS). This is particularly challenging in skull base procedures, where narrow anatomical corridors, frequent occlusions, specular highlights, and visually similar instruments make multi-class detection and 2D pose estimation difficult. We address joint instrument detection and keypoint-based pose estimation from monocular endoscopic videos and introduce MID-POSE, a dual-head architecture that couples a high-resolution HRNetV2p encoder with a class-agnostic dense detection-pose head and a Multi-level Instrument Classification (MIC) head which operates on RoI-aligned multi-level features. To support this task, we construct the PitSurg dataset from 26 clinical procedures, providing seven instrument classes with bounding boxes and detailed 2D keypoints. Using YOLOv8x-pose as our strongest baseline, which in our tasks outperforms YOLO11x-pose, MID-POSE improves Det/Pose $AP_{50-95}$ on PitSurg from 59.4/63.1 to 77.5/78.5 and on the robotic SurgPose dataset from 47.9/61.1 to 62.7/71.4. Qualitative analysis shows that high-resolution features sharpen localisation and keypoint placement, while the RoI classifier reduces misclassifications and spurious background detections, indicating that the proposed architecture and dataset provide an effective basis for robust multi-instrument perception in MIS.

**Keywords:** Minimally Invasive Surgery, Skull Base Surgery, Surgical Instrument Detection, Pose Estimation

---

[*] Contributed equally

## 1. Introduction

Minimally invasive surgery (MIS) has become the standard for many procedures, with surgeons operating through narrow anatomical corridors using elongated instruments visualised by an endoscope (Jeganathan et al., 2025). Joint detection and 2D pose estimation of multiple instruments offer a compact scene representation, enabling geometric reasoning and semantic understanding, which are essential for downstream applications like intraoperative guidance, context-aware assistance, workflow analysis, and skill assessment (Das et al., 2024, 2025). However, accurate instrument perception in MIS remains challenging because instruments are frequently occluded, affected by specular highlights, blood contamination, and motion blur, and can appear visually similar within a confined field of view. Additional complications arise from domain shifts between patients, limited dataset sizes, and expensive manual annotations. These factors make multi-class detection and keypoint estimation much more difficult than generic object detection or human pose estimation in natural images (Maji et al., 2022; Xu et al., 2022).

Deep learning has delivered powerful architectures for object detection and keypoint-based pose estimation (Ren et al., 2015; Newell et al., 2016; Xiao et al., 2018; Chen et al., 2018; Cheng et al., 2020; Xu et al., 2022; Maji et al., 2022). Two-stage *top-down* pipelines typically combine a generic instance detector such as Faster R-CNN (Ren et al., 2015) with a single-instance pose network (Newell et al., 2016; Xiao et al., 2018; Chen et al., 2018; Xu et al., 2022), whereas *bottom-up* methods localise all keypoints jointly and then group them into instances (Cao et al., 2017; Newell et al., 2017; Papandreou et al., 2018; Cheng et al., 2020). More recently, one-stage dense predictors such as YOLO-pose (Maji et al., 2022) jointly output boxes, classes, and keypoints in a single head, improving robustness in crowded scenes. Most of these approaches rely on backbones that repeatedly downsample the input and then attempt to recover spatial detail, whereas high-resolution representations are crucial for localisation-sensitive tasks (Sun et al., 2019).

These generic architectures have increasingly been adapted to surgical instruments. On robotic MIS, Wu et al. (Wu et al., 2025) benchmark YOLOv8x-pose, ViTPose, and DeepLabCut on the SurgPose dataset, showing that human-pose architectures can be transferred to articulated tools. For manual laparoscopy, teams in the PhaKIR challenge (Rueckert et al., 2025) extend YOLOv8-based detectors to predict per-instrument keypoints, including strategies for uncertainty estimation and handling a variable number of keypoints per class. Other works target 6D instrument pose from monocular images using one-stage regression (Yoshimura et al., 2020) or two-stage pipelines that combine YOLO-based detection with crop-based pose networks (Spektor et al., 2024). These studies demonstrate that YOLO-style one-stage detector–pose architectures are strong baselines for surgical instruments and that high-resolution encoders further benefit keypoint localisation. However, support for *truly multi-class, multi-instrument 2D pose estimation from monocular endoscopic views* remains limited, particularly in complex skull base surgery.

Dataset availability is a further bottleneck. SurgPose (Wu et al., 2025) provides articulated 2D keypoints for six robotic instrument types in stereo MIS, and PhaKIR (Rueckert et al., 2025) offers 2D keypoints for 19 laparoscopic instruments. ROBUST-MIPS (Han et al., 2025) and ART-Net (Hasan et al., 2021) focus on tip and shaft representations in abdominal or robotic procedures. These resources provide valuable benchmarks for robotic

and abdominal laparoscopy, but they do not cover monocular endoscopic pituitary surgery, a setting with distinctive visual conditions affecting instruments, such as frequent occlusions, border truncation, extreme perspectives, blur, and specular highlights. To the best of our knowledge, there is no public dataset for endoscopic pituitary surgery that combines multi-class instrument labels with instrument-specific 2D keypoint annotations for the sellar phase (Marcus et al., 2021).

This work addresses both methodological and dataset gaps by proposing MID-POSE (Multi-Instrument Detection and Pose Estimation in Endoscopic surgery), a dual-head architecture for multi-class surgical instrument detection and 2D keypoint pose estimation in MIS, and by constructing a new dataset for endoscopic pituitary surgery. MID-POSE combines a high-resolution HRNet (Sun et al., 2019) encoder with a class-agnostic dense detection-pose head in the style of YOLOv8-pose (Maji et al., 2022) and a proposed Multi-level Instrument Classification (MIC) head which operates on RoI-aligned multi-level features. We evaluate the approach on both the new PitSurg dataset for manual endoscopic pituitary surgery and on the robotic SurgPose benchmark (Wu et al., 2025), allowing us to assess performance across manual and robotic MIS scenarios. The main contributions of this work are:

- **PitSurg: a dataset for endoscopic pituitary surgery**, comprising monocular intraoperative images from 26 procedures with seven instrument types annotated by bounding boxes and detailed, class-specific 2D keypoints under frequent occlusion, truncation, and class imbalance.
- **MID-POSE: a dual-head architecture for multi-class instrument detection and 2D keypoint pose estimation**, which builds on HRNetV2p features with a class-agnostic dense detection-pose head and a MIC head, and incorporates a quality-aware instrumentness objective together with an extended keypoint visibility scheme.
- **A benchmark** for joint detection and 2D pose estimation in both manual and robotic endoscopic MIS scenarios, providing reference results with representative YOLO-style and HRNet-style designs as a common reference point under challenging intraoperative conditions.

## 2. Method: MID-POSE for Instrument Detection and Pose

We propose MID-POSE, a dual-head architecture for multi-class surgical instrument detection and 2D keypoint pose estimation in minimally invasive surgery. The model combines a high-resolution encoder, a class-agnostic dense detection-pose head, and a MIC head operating on RoI-aligned multi-level features (Fig. 1).

### 2.1. Architecture Overview

**Encoder and feature pyramid** We adopt HRNetV2p–W32 as the encoder (Sun et al., 2019) as it maintains a high-resolution stream fused with lower-resolution branches, yielding semantically rich, spatially precise features that have proven effective for localization-sensitive dense prediction tasks. The encoder outputs a three-level feature pyramid ($P_3, P_4$, and $P_5$) with strides 8, 16, and 32. These feature maps preserve fine spatial detail while progressively enriching semantics, and are shared between the dense and the MIC heads.

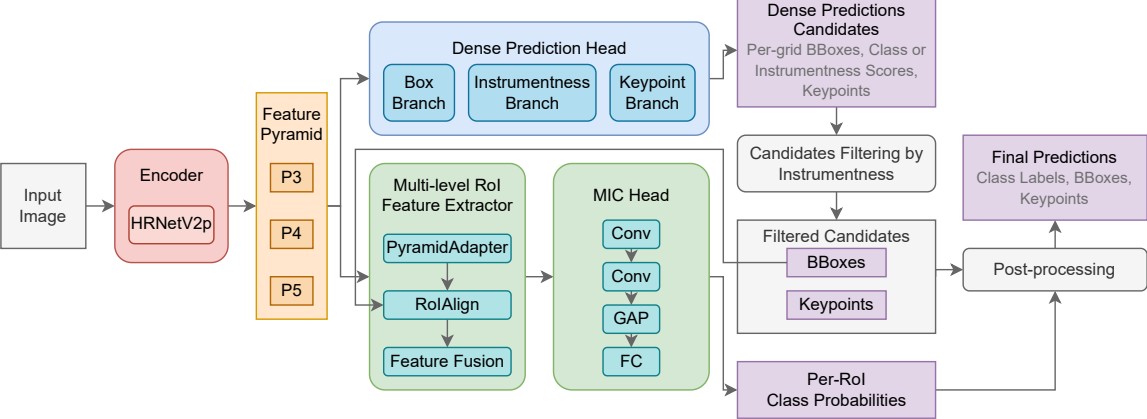

Figure 1: Overview of the proposed MID-POSE architecture for instrument detection and 2D keypoint estimation. Input images are encoded using an HRNetV2p encoder to produce a three-level feature pyramid (P3–P5). A class-agnostic dense prediction head attached to the pyramid outputs per-grid bounding boxes, instrumentness scores, and keypoint candidates that are then filtered by instrumentness. Filtered candidates are then used to extract multi-level RoI features from P3–P5 via a PyramidAdapter and RoIAlign, followed by a MIC head to predict per-RoI instrument class probabilities. These are finally combined with the dense predictions during post-processing to obtain the instrument detections and 2D keypoints.

**Class-Agnostic Dense Head** On top of $P_3$–$P_5$, we adopt a YOLOv8-Pose style head (Maji et al., 2022) for joint binary detection and 2D pose estimation in a class-agnostic manner. At every spatial location $a$ on each pyramid level, the head predicts: (i) an instrumentness score $p_a = \sigma(z_a)$, where $\sigma(\cdot)$ denotes the sigmoid function and $z_a$ is the corresponding raw logit, (ii) a bounding box $b_a = (x_{a,1}, y_{a,1}, x_{a,2}, y_{a,2})$, and (iii) $K$ keypoints $k_{a,i} = (x_{a,i}, y_{a,i}, p_{v,a,i})$, $i = 1, \ldots, K$, with 2D coordinates $(x_{a,i}, y_{a,i})$ and visibility probabilities $p_{v,a,i} \in [0, 1]$. Predictions with instrumentness below a fixed threshold $\tau$ are discarded. The remaining candidates are used both as final detection/pose outputs and as proposals for the multi-level RoI classification head.

**Multi-scale RoI Feature Extractor** To assign instrument categories, we introduce a multi-level RoI feature extractor operating on $P_3$–$P_5$. The bounding boxes of the filtered candidates are treated as regions of interest (RoIs). A PyramidAdapter first maps each feature map $P_\ell$ to a common channel dimension $C_r$ via a $1 \times 1$ convolution:

$$P_3, P_4, P_5 \to \tilde{P}_3, \tilde{P}_4, \tilde{P}_5 \in \mathbb{R}^{C_r \times H_\ell \times W_\ell}$$

, while preserving their original spatial resolutions. For every RoI, RoIAlign is applied independently to $\tilde{P}_3, \tilde{P}_4, \tilde{P}_5$, yielding tensors of size $C_r \times H_{\mathrm{roi}} \times W_{\mathrm{roi}}$ at each level. These three tensors are concatenated along the channel dimension to form a multi-scale descriptor of

shape $3 \times C_r \times H_{\text{roi}} \times W_{\text{roi}}$ that combines detailed local information from $P_3$ with increasingly contextual features from $P_4$ and $P_5$.

The concatenated tensor is passed through a feature fusion module consisting of a $1 \times 1$ convolution, group normalization, and a SiLU activation, which reduces the channels back to $C_r$ and learns to mix information across levels. The output is a fused RoI feature map for each candidate bounding box.

**MIC head** The MIC head operates on the fused RoI features and predicts a discrete probability distribution over the instrument categories. It consists of two convolutional refinement blocks (Conv), each composed of a $3 \times 3$ convolution with stride 1 and padding 1, followed by group normalization and a SiLU activation; these blocks refine the local RoI features while preserving the $H \times W$ spatial resolution of the feature map. Global average pooling (GAP) is then applied over the spatial dimensions to obtain a $C_r$-dimensional descriptor for each RoI. This descriptor is passed through a fully connected block (FC) with two layers, where the first is a hidden linear layer with ReLU activation and dropout and the second is an output layer that produces eight logits corresponding to the seven pituitary instrument types and a background class.

## 2.2. Loss Functions

**Instrumentness loss** Let $y_a \in [0, 1]$ be a soft target that reflects how well the predicted instrument at location $a$ matches its ground truth, increasing with both the instrumentness confidence and the IoU between the predicted and ground-truth bounding boxes, and let $p_a$ denote the corresponding instrumentness probability. A quality-aware focal loss inspired by VarifocalNet (Zhang et al., 2021) is used to train this branch. The per-location instrumentness loss is defined as:

$$\mathcal{L}_{\text{inst}}(p_a, y_a) = w(p_a, y_a) \left[ -y_a \log p_a - (1 - y_a) \log(1 - p_a) \right] \tag{1}$$

, with

$$w(p_a, y_a) = \mathbb{1}[y_a > 0] \, y_a + \mathbb{1}[y_a = 0] \, \alpha \, p_a^{\gamma} \tag{2}$$

, where $\alpha$ and $\gamma$ control the balance between positive and negative locations and the degree of focusing on hard negatives. Because the dense head produces predictions at every spatial location while only a small top-$k$ subset is assigned as positives, this quality-aware focal reweighting prevents the loss from being dominated by easy background locations and encourages the model to focus on well-localized positives and hard negatives.

**Bounding box loss** The bounding box loss $\mathcal{L}_{\text{box}}$ follows the default YOLOv8-Pose formulation (Maji et al., 2022) and is computed only for positive grid locations ($y_a > 0$).

**Keypoint loss** The keypoint loss $\mathcal{L}_{\text{kpt}}$ follows the YOLOv8-pose formulation (Maji et al., 2022) and is computed only for positive grid locations ($y_a > 0$). It is decomposed into an Object Keypoint Similarity (OKS) coordinate term and a visibility term:

$$\mathcal{L}_{\text{kpt}} = \lambda_{\text{loc}} \mathcal{L}_{\text{OKS}} + \lambda_{\text{vis}} \mathcal{L}_{\text{vis}}. \tag{3}$$

We extend the visibility label $v_j$ for keypoint $j$ to the range $v_j \in \{-1, 0, 1, 2\}$, where $v_j = -1$ denotes an unlabeled keypoint. For the coordinate term, only keypoints with $v_j > 0$ are treated as present:

$$\mathcal{L}_{\text{OKS}}(j) = \begin{cases} 0, & v_j \leq 0 \\ 1 - \exp\left(-\dfrac{\|\hat{k}_j - k_j\|_2^2}{2\,A\,(2\sigma_j)^2 + \varepsilon}\right), & v_j > 0 \end{cases} \quad (4)$$

, where $\hat{\mathbf{k}}_j$ and $\mathbf{k}_j$ are the predicted and ground-truth keypoint locations, $A$ is the ground-truth box area, and $\sigma_j$ is a keypoint-specific tolerance. For the visibility term, a binary target $t_j = \mathbb{K}[v_j > 0]$ is defined for labeled keypoints with $v_j \geq 0$, and the per-keypoint loss is:

$$\mathcal{L}_{\text{vis}}(j) = \begin{cases} 0, & v_j = -1 \\ -\Big[t_j \log \sigma(\hat{v}_j) + (1 - t_j)\log\big(1 - \sigma(\hat{v}_j)\big)\Big], & v_j \geq 0 \end{cases} \quad (5)$$

, where $\sigma(\cdot)$ denotes the sigmoid. This extension allows instances without keypoint annotations to contribute to detection while excluding their unlabeled keypoints from the pose supervision.

**MIC classification loss** For each RoI $r$, the MIC head outputs logits $s_r \in \mathbb{R}^8$ over seven instrument classes plus background. Let $y_r \in \{0, \ldots, 7\}$ denote the ground-truth label for RoI $r$ and $p_{r,c}$ the softmax probability for class $c$. The MIC loss is the standard cross-entropy:

$$\mathcal{L}_{\text{mic}} = \frac{1}{N_{\text{mic}}} \sum_{r=1}^{N_{\text{mic}}} \big[-\log p_{r,y_r}\big] \quad (6)$$

, where $N_{\text{mic}}$ is the number of RoIs in the batch. This penalizes low predicted probability for the ground-truth class and drives the RoI head to discriminate between the seven instrument categories and background.

**Total loss** The network is trained end-to-end by combining the dense detection-pose losses from the class-agnostic head with the categorical loss from the MIC head. For each batch, the total loss is:

$$\mathcal{L}_{\text{total}} = \lambda_{\text{box}} \mathcal{L}_{\text{box}} + \lambda_{\text{inst}} \mathcal{L}_{\text{inst}} + \lambda_{\text{kpt}} \mathcal{L}_{\text{kpt}} + \lambda_{\text{mic}} \mathcal{L}_{\text{mic}} \quad (7)$$

, where $\lambda_{\text{box}}, \lambda_{\text{inst}}, \lambda_{\text{kpt}}$, and $\lambda_{\text{mic}}$ control the relative contributions of localization, instrumentness, pose, and MIC.

## 3. Dataset and Experimental Setup

We evaluate MID-POSE on two complementary datasets: *PitSurg*, a new dataset of manual endoscopic pituitary surgery, and *SurgPose*, a public benchmark of robotic MIS.

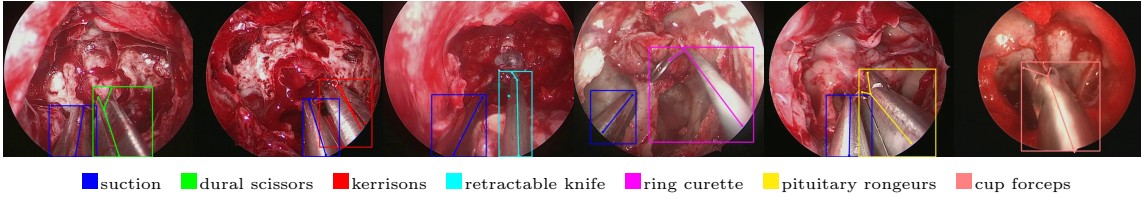

Figure 2: Examples of PitSurg instrument classes with bounding boxes and 2D keypoints.

**PitSurg** dataset is derived from 26 videos of monocular endoscopic pituitary surgery performed at the National Hospital for Neurology and Neurosurgery, London, UK. Videos were captured using a Hopkins Telescope with an AIDA storage system (Karl Storz Endoscopy, UK) at 720p, 24 FPS, and frames were sampled at 1 FPS for annotation (a summary is provided in Appendix A, Table 4). We used only sellar-phase frames (Marcus et al., 2021) that contain one or two visible instruments, and split the data at the procedure level so that all frames from a given surgery appear in either the training or validation set, avoiding patient-level leakage. Seven instrument types are annotated, as illustrated in Fig. 2, namely Suction, Dural Scissors, Kerrisons, Retractable Knife, Ring Curette, Pituitary Rongeurs, and Cup Forceps. Each instance has a bounding box, class label, and 2D keypoints with class-specific layouts, with suction having 2 keypoints, ring curette 3, retractable knife, dural scissors, pituitary rongeurs, and cup forceps 4, and Kerrisons 5. PitSurg reflects real intraoperative imaging conditions, in which annotated instruments are affected by frequent occlusion, border truncation, motion blur, extreme perspectives, and specular highlights. Consequently, not all keypoints are visible in every frame. The prevalence of these instrument-level challenging conditions is summarised in Appendix A (Figure 5). Only suction can appear together with any other instrument. Because clinically verified cup forceps annotations are scarce in the sellar phase, we augment the training set with 376 additional cup forceps instances annotated only with bounding boxes and class labels, extracted from non-sellar segments of the same procedures, while keeping the validation set restricted to sellar-phase frames. The training split contains 1,042 suction, 351 dural scissors, 495 Kerrisons, 331 retractable knife, 312 ring curette, 301 pituitary rongeurs, and 487 cup forceps instances, and the validation split contains 201, 80, 114, 95, 91, 51, and 83 instances, respectively.

**SurgPose** dataset (Wu et al., 2025) contains stereo endoscopic videos acquired with a da Vinci surgical system. Following the official protocol, we use only the left-view images and adopt the provided split, using trajectories 0–19 for training and 20–33 for validation. Each frame contains two articulated robotic instruments annotated with bounding boxes and five 2D keypoints per tool across six instrument types: Large Needle Driver (LND), Mega Needle Driver (MND), MicroForceps, Curved Scissor, DeBakey Forceps, and Prograsp Forceps. In this dataset, instruments are fully visible in all frames, without occlusion, blur, specular highlights or end-effector border truncation.

**Five model variants** are considered in our experiments, namely YOLOv8x-pose (Maji et al., 2022) and YOLO11x-pose (Ultralytics, 2024); a YOLOv8x-pose+MIC, where the YOLOv8x-pose is augmented with the proposed MIC head to form a dual-head architecture;

an HRNetV2p-pose model, which uses an HRNetV2p encoder feeding a YOLO–style dense detection-pose head; and the proposed MID-POSE architecture.

**Training protocol** All variants are implemented in PyTorch using Ultralytics YOLO (Jocher et al., 2023), with custom extensions for MID-POSE, initialised from COCO-pretrained checkpoints and trained on a single NVIDIA GeForce RTX 4090 GPU. For both datasets, all models use the same augmentations. Images are resized to $640 \times 640$ and augmented with random rotations (up to $\pm20°$), translations (up to $10\%$), isotropic scaling in $[0.5, 1.5]$, horizontal flips (probability 0.5), and mild photometric jitter in hue, saturation, and brightness.

Optimisation uses stochastic gradient descent (SGD) with a learning rate of 0.01, momentum 0.9, and weight decay $5 \times 10^{-4}$. Models are trained for 80 epochs on PitSurg and 50 epochs on SurgPose, with a batch size of 16 for YOLOv8x-pose, YOLO11x-pose, and YOLOv8x-pose+MIC, and 8 for HRNetV2p-Pose and MID-POSE. The batch size for models utilizing the HRNetV2p encoder was reduced to accommodate the higher memory footprint required by its high-resolution feature maps. YOLOv8x-pose+MIC and MID-POSE use two-stage training: first the encoder and class-agnostic dense head are trained as a binary detector–pose model, then the full dual-head architecture is fine-tuned while RoIs are constructed online from dense predictions. For each ground-truth instrument we keep the three predictions with the highest $p_a$ as positive RoIs and three hard negatives as the highest-$p_a$ predictions with zero IoU to all ground-truth boxes, labelling them with the corresponding instrument class or background. At inference, dual-head models filter dense predictions with an instrumentness threshold $\tau = 0.3$, keeping only predictions with $p_a \geq \tau$ as candidates for MIC.

Following the baseline YOLOv8x-pose implementation by ultralytics (Jocher et al., 2023), the default values of weighting parameters (Eq. (7)), $\lambda_{\mathrm{box}} = 7.5$, $\lambda_{\mathrm{kpt}} = 12.0$ and $\lambda_{\mathrm{inst}} = 1.0$ are used in the loss function for all variants on both datasets. On PitSurg we use $\lambda_{\mathrm{cls}} = 1.0$ for the single-head variants (YOLOv8x-Pose, YOLO11x-Pose, HRNetV2p-Pose), where $\mathcal{L}_{\mathrm{cls}}$ is the multi-class counterpart of $\mathcal{L}_{\mathrm{inst}}$, and we use $\lambda_{\mathrm{mic}} = 10.0$ for the dual-head variants (YOLOv8x-pose+MIC and MID-POSE). On SurgPose, single-head models use $\lambda_{\mathrm{cls}} = 40.0$, and dual-head models use $\lambda_{\mathrm{mic}} = 17.0$. In all cases, the $\lambda$ values are set according to the relative scales and difficulty of the underlying tasks so that each loss contributes in a balanced way. We also note that globally scaling these weighting parameters is effectively equivalent to adjusting the learning rate for those terms.

**Evaluation metrics** We report detection and pose performance using average precision over thresholds from 0.5 to 0.95 for IoU (Det $mAP_{50-95}$) and OKS (Pose $mAP_{50-95}$), given per class and as a class-averaged overall score in percentage. For qualitative examples, we show the mean IoU and OKS per image, assigning IoU $= 0$ and OKS $= 0$ to false negatives and computing these scores only for true positives.

## 4. Results and Discussion

We compare dense single-head architectures (YOLOv8x-pose, YOLO11x-pose, HRNetV2p-pose) and dual-head variants (YOLOv8x-pose+MIC and MID-POSE) on the PitSurg and SurgPose datasets. Performance is measured using Det/Pose $AP_{50-95}$ per class and overall

Table 1: Detection and pose $AP_{50-95}$ (in %) on the (a) PitSurg and (b) SurgPose validation sets for different encoder-head combinations. Results are reported per instrument class and as an overall class-averaged AP for detection (Det) and pose (Pose).

| Architecture | (a) PitSurg Dataset - $AP_{50-95}$ (Det/Pose) | | | | | | | |
| --- | --- | --- | --- | --- | --- | --- | --- | --- |
| | Overall | suction | dural_scissors | kerrisons | retractable_knife | ring_curette | pituitary_rongeurs | cup_forceps |
| YOLOv8x-pose (Maji et al., 2022) | 59.4 / 63.1 | 70.6 / 84.5 | 65.1 / 79.5 | 81.6 / 57.5 | 53.5 / 49.6 | 37.7 / 66.4 | 56.3 / 57.6 | 50.8 / 46.3 |
| YOLO11x-pose (Ultralytics, 2024) | 54.8 / 62.0 | 66.5 / 85.2 | 56.7 / 74.2 | 75.9 / 59.1 | 42.2 / 36.7 | 45.1 / 72.9 | 51.2 / 57.1 | 45.7 / 49.1 |
| YOLOv8x-pose+MIC | 59.6 / 64.9 | 74.7 / 87.7 | 57.5 / 73.4 | 76.9 / 67.6 | 57.8 / 54.1 | 48.9 / 76.6 | 50.1 / 44.3 | 51.6 / 51.0 |
| HRNetV2p-pose | 73.3 / 76.5 | 80.0 / **92.5** | 72.1 / **82.3** | 88.2 / 77.4 | 70.6 / 64.2 | 63.2 / 82.4 | 63.1 / 66.1 | **75.6** / 70.4 |
| MID-POSE | **77.5 / 78.5** | **82.9** / 92.3 | **76.1 / 82.3** | **89.6 / 81.2** | **78.7** / 65.6 | **73.6 / 86.8** | **68.3 / 68.2** | 73.3 / **73.2** |

| Architecture | (b) SurgPose Dataset - $AP_{50-95}$ (Det/Pose) | | | | | | |
| --- | --- | --- | --- | --- | --- | --- | --- |
| | Overall | LND | MND | MicroForceps | Scissor | Forceps | Prograsp |
| YOLOv8x-pose (Maji et al., 2022) | 47.9 / 61.1 | 74.2 / 82.0 | 31.4 / 30.6 | 26.4 / 36.2 | 42.4 / 83.9 | 32.7 / 41.4 | **80.2 / 92.6** |
| YOLO11x-pose (Ultralytics, 2024) | 47.2 / 60.9 | 72.3 / 80.8 | 31.3 / 30.4 | 26.0 / 36.2 | 41.8 / 83.8 | 32.2 / 41.5 | 79.8 / 92.6 |
| YOLOv8x-pose+MIC | 59.7 / 68.2 | 77.6 / 79.3 | 61.9 / 54.5 | 57.3 / 66.0 | 54.0 / **95.0** | 39.1 / 43.7 | 68.3 / 70.9 |
| HRNetV2p-pose | 50.3 / 64.9 | 76.0 / **84.7** | 41.9 / 45.9 | 47.6 / 55.6 | 26.9 / 66.0 | 36.5 / 48.8 | 72.9 / 88.3 |
| MID-POSE | **62.7 / 71.4** | **79.5** / 81.2 | **63.4 / 55.9** | **59.5 / 68.5** | **55.2 / 95.0** | **48.8 / 56.0** | 69.6 / 72.1 |

(Table 1). As YOLO11 does not yield consistent gains over YOLOv8x, we use YOLOv8x-pose as the primary baseline and include YOLO11x-pose for completeness.

**Quantitative** Table 1 reports detection and pose $AP_{50-95}$ on the PitSurg and SurgPose datasets, respectively. On PitSurg, the primary performance driver is the encoder: replacing YOLOv8x with HRNetV2p improves baseline $AP_{50-95}$ (Det/Pose) from 59.4% / 63.1% to 73.3% / 76.5%, with particular gains on challenging classes like retractable knife, ring curette and cup forceps. The MIC head adds a further boost, resulting in the best performance of 77.5% / 78.5%. Conversely, on SurgPose, the dual-head design acts as the dominant factor. While the HRNetV2p encoder offers modest gains over the YOLOv8x baseline (47.9% / 61.1%), adding the MIC head to YOLOv8x jumps performance to 59.7% / 68.2%. The proposed MID-POSE architecture achieves the best overall SurgPose results, with 62.7% / 71.4%. To further isolate the contribution of HRNetV2p to spatial localization from the effects of the classification design, we also report class-agnostic performance in Table 7. This indicates a complementary relationship: high-resolution features (HRNetV2p) drive spatial precision, critical for PitSurg, while the MIC head resolves class confusion among similar instrument tips, which is the primary bottleneck in SurgPose.

**Qualitative** Qualitative examples (Fig. 3 and Fig. 4) illustrate the distinct roles of the encoder and the MIC head. Across both datasets, the HRNetV2p encoder enhances localization; it produces bounding boxes that tightly follow instrument shafts and captures occluded parts (e.g., ring curette) where the baseline often over-extends into background tissue. In contrast, the MIC head primarily improves semantic consistency. It suppresses background false positives and corrects mislabeled bounding boxes, specifically in SurgPose, where single-head baselines struggle to distinguish instruments with similar end-effectors. While the encoder improves per-example IoU and OKS through richer spatial features, the MIC head ensures that geometrically similar bounding boxes receive coherent class labels and higher confidence scores, driving the AP gains through true positive recovery rather than further spatial refinement.

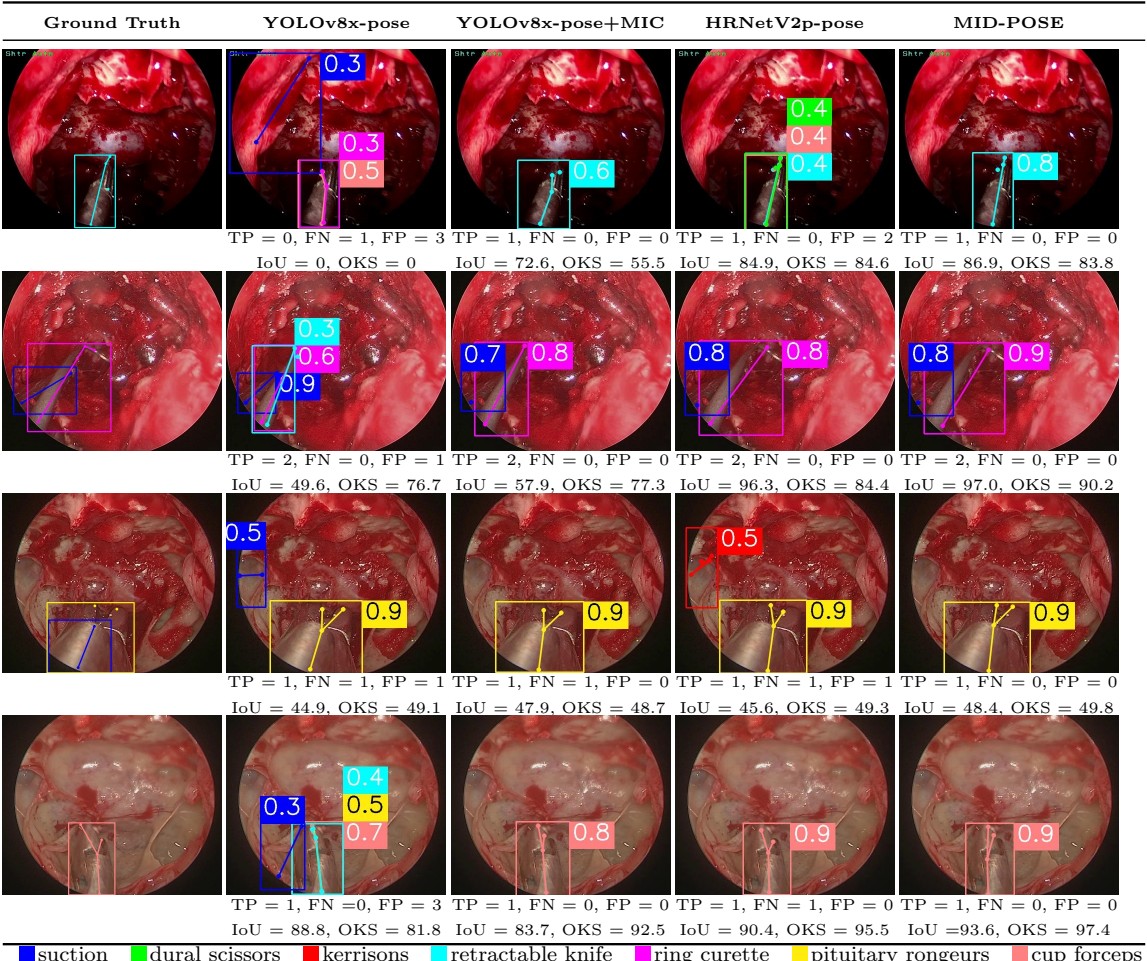

Figure 3: Qualitative PitSurg examples with ground truth and predictions. TP/FN/FP counts and mean IoU/OKS are shown below each prediction.

**Sensitivity to weighting parameter $\lambda_{\mathrm{mic}}$** The sensitivity study shown Table 2 investigates the influence of weighting factor $\lambda_{\mathrm{mic}}$ on detection and pose performance of dual-head models on both PitSurg and SurgPose datasets. One can observe that lower $\lambda_{\mathrm{mic}}$ values result in higher performance on PitSurg, while higher $\lambda_{\mathrm{mic}}$ values favour SurgPose. Importantly, across the tested range $\lambda_{\mathrm{mic}} \in [10.0, 17.0]$, MID-POSE consistently outperforms YOLOv8x-pose+MIC on both datasets, indicating that our qualitative conclusions are not sensitive to a narrowly tuned choice of loss weight.

**Sensitivity to instrumentness threshold $\tau$** Table 3 reports the False Positives (FP) and False Negatives (FN) trade-off under different instrumentness thresholds $\tau \in \{0.1, 0.3, 0.5\}$, together with success rates defined as $\Pr(\mathrm{IoU} \geq 0.5)$ for detection and $\Pr(\mathrm{OKS} \geq 0.5)$ for pose. Across both PitSurg and SurgPose, $\tau = 0.3$ provides a practical operating point, as it yields a large reduction in false positives (i.e., "ghost" detections) while incurring

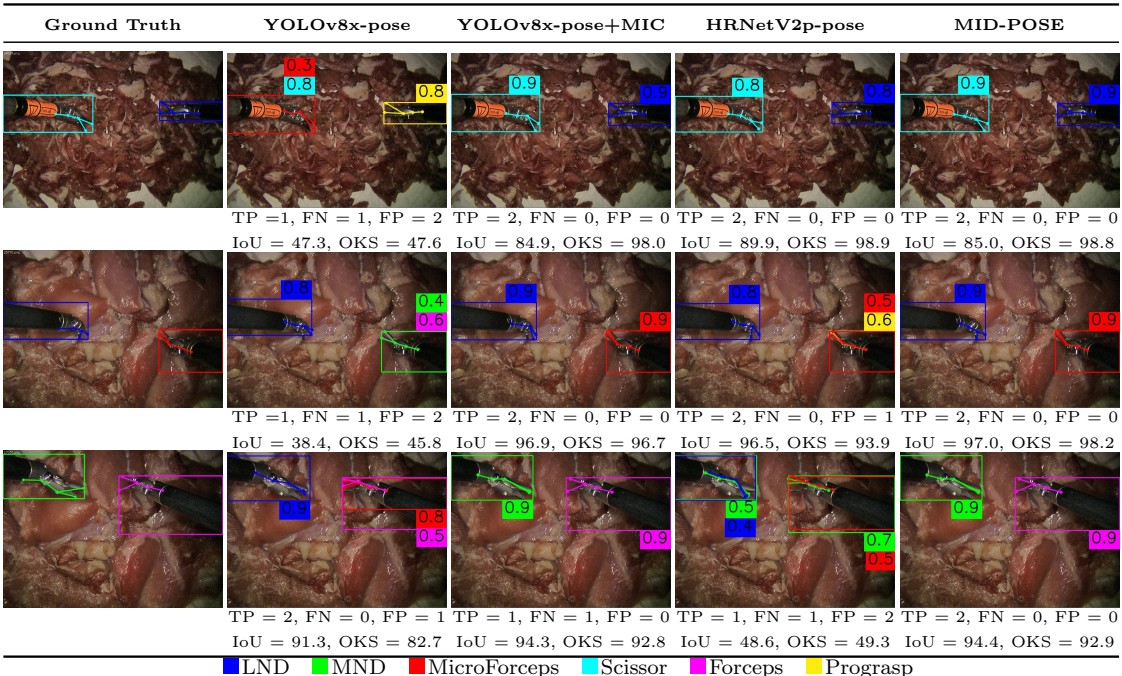

Figure 4: Qualitative SurgPose examples with ground truth and predictions. TP/FN/FP counts and mean IoU/OKS per image are reported below each prediction.

Table 2: Sensitivity to weighting parameter $\lambda_{\mathrm{mic}}$.

| Architecture ↓ / $\lambda_{\mathrm{mic}}$ → | PitSurg Dataset - $mAP_{50-95}$ (Det/Pose) | | | | SurgPose Dataset - $mAP_{50-95}$ (Det/Pose) | | | |
|---|---|---|---|---|---|---|---|---|
| | 10.0 | 12.5 | 15.0 | 17.0 | 10.0 | 12.5 | 15.0 | 17.0 |
| YOLOv8x-pose+MIC | 59.6/64.9 | 59.4/64.8 | 58.8/64.2 | 58.1/63.6 | 56.4/64.8 | 58.7/67.2 | 59.1/67.4 | 59.7/68.2 |
| MID-POSE | 77.5/78.5 | 77.2/78.2 | 76.6/77.4 | 75.9/76.8 | 59.3/68.4 | 60.9/69.8 | 62.1/70.9 | 62.7/71.4 |

only a marginal increase in false negatives. Additionally, MID-POSE demonstrates reduced threshold sensitivity over YOLOv8x-pose. Varying $\tau$ from 0.1 to 0.5 in MID-POSE maintains success rates above 90% on PitSurg and above 79% on SurgPose, while YOLOv8x-pose is much more sensitive, with success rates decreasing from above 90% to around 60%. This suggests MID-POSE delivers more consistent performance over threshold shifts for deployment in distractor-heavy surgical scenes.

**Discussion** Overall, PitSurg and SurgPose highlight complementary strengths of the proposed architecture: HRNetV2p mainly improves spatial precision, whereas the MIC head addresses fine-grained semantic ambiguities between visually similar instruments. Despite the higher training memory footprint of HRNetV2p, the MIC module itself is lightweight, containing 2.18M parameters compared with 7.38M in the baseline YOLOv8x-pose classification head. This suggests that the semantic gains are driven by object-centric feature extraction from multi-level RoIs. Our systematic failure analysis (Appendix B) quantifies these improvements across condition categories in both PitSurg and SurgPose, showing that

Table 3: Sensitivity to instrumentness threshold $\tau$.

| $\tau$ | Metric | PitSurg | | SurgPose | |
|---|---|---|---|---|---|
| | | YOLOv8x-pose | MID-POSE | YOLOv8x-pose | MID-POSE |
| 0.1 | FN / FP | 21 / 1519 | 18 / 68 | 13 / 51499 | 17 / 28 |
| | IoU / OKS* | 94.8 / 96.2 | 94.3 / 93.9 | 94.7 / 94.7 | 84.7 / 84.7 |
| 0.3 | FN / FP | 26 / 1054 | 21 / 27 | 45 / 14171 | 18 / 25 |
| | IoU / OKS* | 85.7 / 87.0 | 92.5 / 92.2 | 79.6 / 79.5 | 83.0 / 83.0 |
| 0.5 | FN / FP | 81 / 214 | 42 / 22 | 815 / 2906 | 35 / 21 |
| | IoU / OKS* | 60.5 / 71.4 | 90.7 / 90.1 | 69.5 / 69.5 | 79.8 / 79.8 |

*Detection / pose success rate, defined as $Pr(IoU \geq 0.5)$ / $Pr(OKS \geq 0.5)$, in %.

the observed gains are not driven only by easier instances. On PitSurg (Table 5), MID-POSE eliminates failures in the Blur, Specular Highlights, and Clean categories, and substantially improves Partial Occlusion (6.75% → 0.78%). The most difficult categories for YOLOv8x–pose are Heavy Occlusion (40.91%), Border Truncation (22.50%), and Extreme Perspectives (19.10%). MID-POSE reduces these failure rates to 10.00%, 6.25%, and 13.48%, respectively. Figure 6 further audits these categories through systematically selected worst-case examples. On SurgPose (Table 6), YOLOv8x-pose fails in 30.10%, 43.60%, and 20.70% of instances under the Semantic Ambiguity, Extreme Perspectives, and Clean categories, respectively, whereas MID-POSE reduces these to 7.35%, 10.00%, and 0.36%. Figure 7 provides corresponding worst-case qualitative evidence for representative failure categories in SurgPose.

## 5. Conclusion

We presented MID-POSE, a dual-head architecture for multi-class surgical instrument detection and 2D keypoint pose estimation in minimally invasive surgery, together with the PitSurg dataset of endoscopic pituitary procedures with class-specific 2D keypoint annotations. By combining a high-resolution HRNetV2p encoder, a class-agnostic dense detection-pose head, and a MIC head operating on RoI-aligned features, MID-POSE consistently improves Det/Pose $AP_{50-95}$ over strong YOLOv8x-pose baselines on both PitSurg and the robotic SurgPose benchmark, with particularly large gains for visually similar instruments and under occlusion. Qualitative results confirm that high-resolution features mainly enhance localisation and pose accuracy, whereas the MIC head resolves fine-grained class ambiguities and suppresses background false positives. Future work will explore spatio-temporal modelling of instrument appearance and motion across video frames to improve robustness, weaker forms of supervision to reduce reliance on dense annotations, and integration into real-time surgical assistance systems.

## Acknowledgments

This work was supported in whole, or part, by the UCL Hawkes Institute – formerly WEISS (203145/Z/16/Z); the EPSRC (EP/Y01958X/1, EP/W00805X/1, EP/Z534754/1); UKRI [UKRI145] grants; DZK is supported by the NIHR Academic Clinical Fellowship; HJM is supported by the WEISS [NS/A000050/1] and by the NIHR Biomedical Research Centre at UCL, has shares in and is employed by Panda Surgical Ltd.; DS is supported by the Department of Science, Innovation and Technology (DSIT), and the Royal Academy of Engineering under the Chair in Emerging Technologies programme. For the purpose of open access, the author has applied a Creative Commons Attribution (CC BY) licence to any Author Accepted Manuscript version arising.

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

## Appendix A. Additional Details on the PitSurg Dataset

PitSurg acquisition details and dataset size are presented in Table 4, while Figure 5 summarises the distribution of instrument-level visual condition categories.

Table 4: PitSurg dataset specifications. Summary of acquisition settings and dataset size.

| Acquisition details | |
| --- | --- |
| Endoscope | Hopkins Telescope with AIDA storage |
| Manufacturer | Karl Storz Endoscopy, UK |
| Native Resolution | 720p |
| Native Frame Rate | 24 FPS |
| Sampling Rate | 1 FPS for annotation |
| **Size of annotated data** | |
| Procedures | 26 clinical videos |
| Instrument Types | 7 classes |
| Total Instances | 4034 instances (3319 Train / 715 Val) |

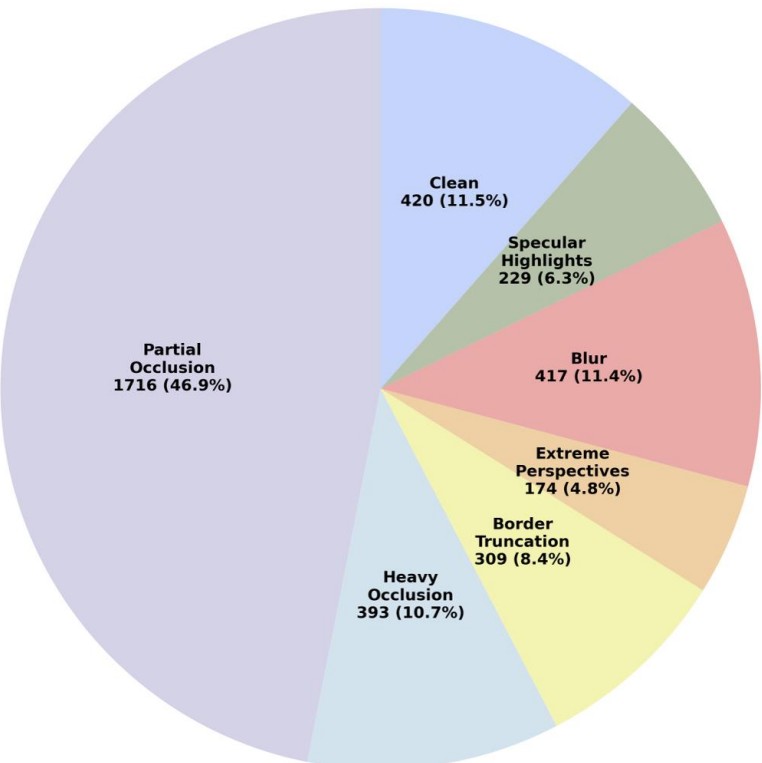

Figure 5: Distribution of instrument-level visual condition categories in PitSurg. Categories include clean instances and degraded regimes, namely partial/heavy occlusion, border truncation, blur, specular highlights, and extreme viewpoints; percentages and counts are shown for each category.

## Appendix B. Systematic Failure Analysis

Tables 5 and 6 report instance-level failure rates of YOLOv8x-pose and MID-POSE across different visual condition categories. Table 5 summarizes results on PitSurg, while Table 6 reports the analogous breakdown on SurgPose. The GT Count ($n$) column indicates the number of ground-truth instrument instances in each category in the validation set.

**Failure rate** We compute failure rates per category using one-to-one matching between predictions and ground-truth instances at IoU $\geq 0.5$. A ground-truth instance is counted as a failure if it is unmatched or if it is matched but the predicted keypoints do not meet the pose criterion (OKS $< 0.5$). The failure rate for a category is the fraction of ground-truth instances in that category that fail under this definition.

**Observations** On PitSurg (Table 5), MID-POSE eliminates failures in the Blur, Specular highlights, and Clean categories (0.00%), and improves Partial Occlusion (6.75% $\rightarrow$ 0.78%). YOLOv8x-pose fails most under Heavy Occlusion (40.91%), Border Truncation

(22.50%), and Extreme Perspectives (19.10%). MID-POSE reduces these to 10.00%, 6.25%, and 13.48%, respectively. Remaining failures are dominated by complete overlap, severe low-contrast truncation, and rare-viewpoint confusion (pituitary rongeurs vs cup forceps); worst-case examples are shown in Figure 6. On SurgPose (Table 6), which provides a complementary benchmark with different visual characteristics, YOLOv8x-pose fails in 30.10%, 43.60%, and 20.70% of instances under the Semantic Ambiguity, Extreme Perspectives, and Clean categories, respectively, while MID-POSE reduces these to 7.35%, 10.00%, and 0.36%. This consistent reduction across categories supports the stability of MID-POSE's robustness gains. Remaining errors are dominated by residual class confusion under rare viewpoints, especially between LND and Forceps (Fig. 7).

Table 5: Failure rates of YOLOv8x-pose and MID-POSE across instrument-level visual condition categories in the PitSurg validation set.

| Condition Category | GT Count ($n$) | YOLOv8x-pose Failure Rate | MID-POSE Failure Rate |
|---|---|---|---|
| Partial Occlusion | 385 | 6.75% | 0.78% |
| Heavy Occlusion | 110 | 40.91% | 10.00% |
| Border Truncation | 80 | 22.50% | 6.25% |
| Extreme Perspectives | 89 | 19.10% | 13.48% |
| Blur | 16 | 12.50% | 0.00% |
| Specular highlights | 12 | 8.33% | 0.00% |
| Clean | 23 | 4.35% | 0.00% |
| TOTAL | 715 | 15.20% | 4.67% |

Table 6: Failure rates of YOLOv8x-pose and MID-POSE across instrument-level visual condition categories in the SurgPose validation set.

| Condition Category | GT Count ($n$) | YOLOv8x-pose Failure Rate | MID-POSE Failure Rate |
|---|---|---|---|
| Semantic Ambiguity | 8,200 | 30.10% | 7.35% |
| Extreme Perspectives | 5,741 | 43.60% | 10.00% |
| Clean | 10,371 | 20.70% | 0.36% |
| TOTAL | 24,312 | 29.30% | 5.00% |

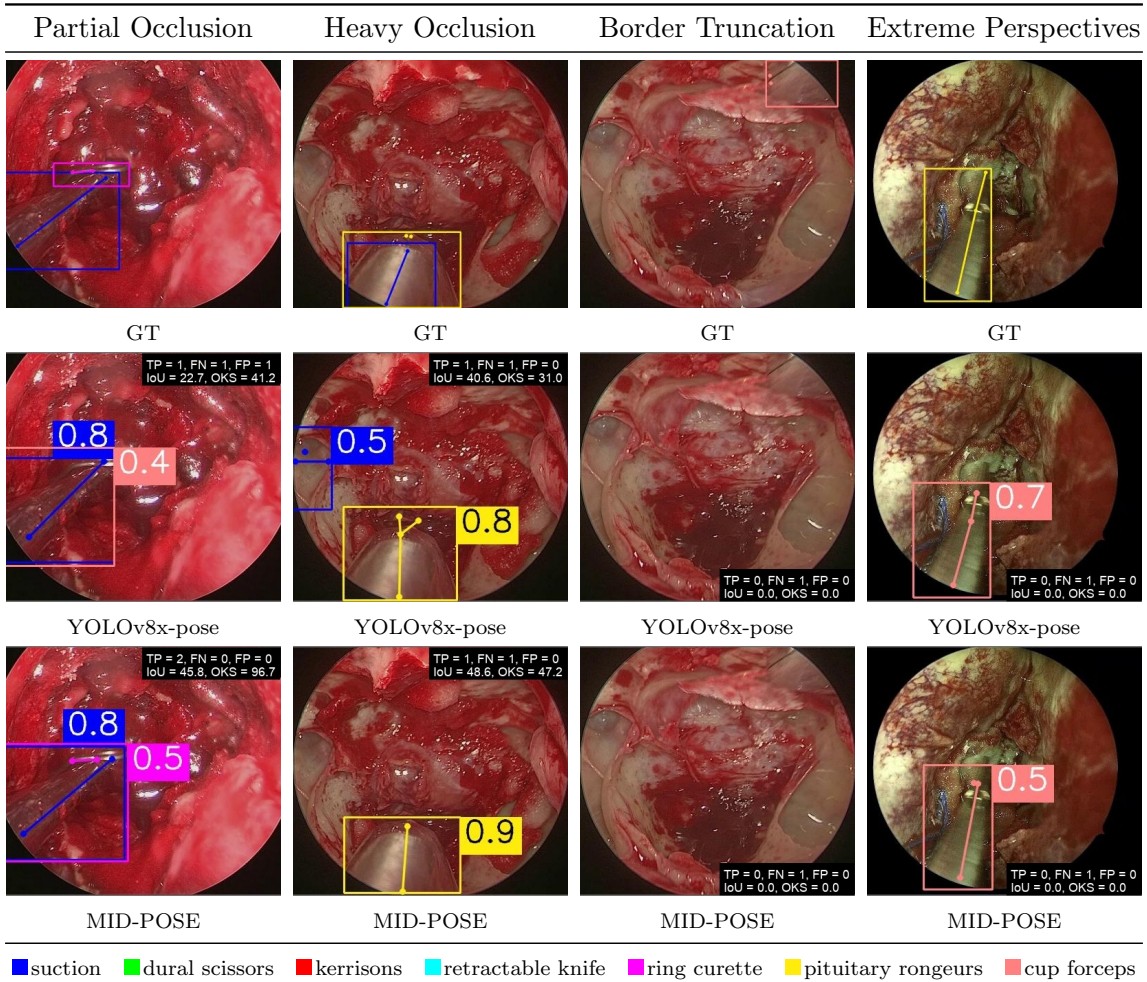

Figure 6: Systematic failure audit in PitSurg across instrument-level visual condition categories. Ground-truth (GT) overlays are shown alongside predictions from YOLOv8x-pose and MID-POSE for representative cases of partial occlusion, heavy occlusion, border truncation, and extreme perspectives.

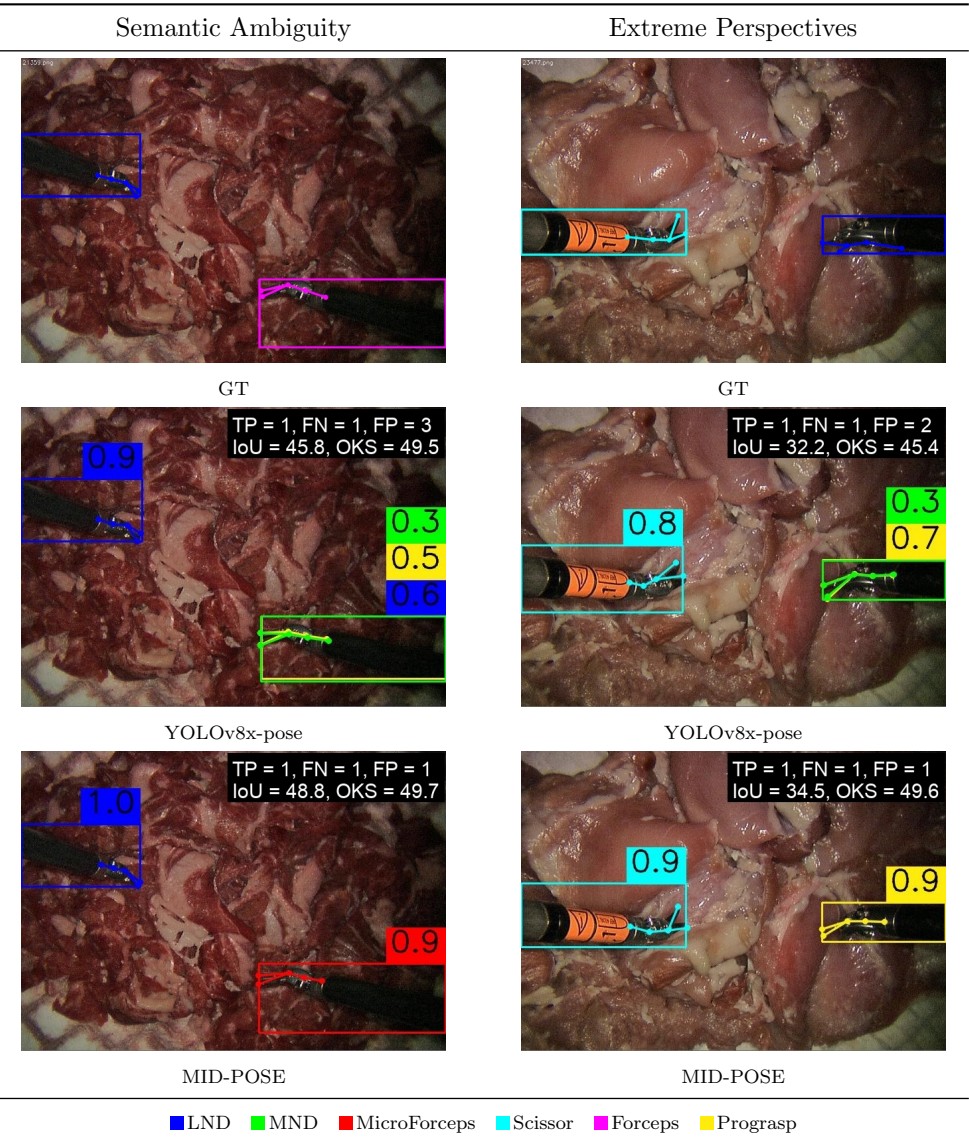

Figure 7: Systematic failure audit in SurgPose for representative condition categories. Ground-truth (GT) overlays are shown alongside predictions from YOLOv8x-pose and MID-POSE for representative cases of semantic ambiguity and extreme perspectives.

## Appendix C. Class-Agnostic Performance

Table 7 reports class-agnostic results, which isolate the contribution of high-resolution feature maps from the effects of the classification design. In the PitSurg dataset, replacing the YOLOv8x encoder with HRNetV2p results in a substantial gain in detection and pose mAP from 46.3% / 67.5% to 84.6% / 84.5%, confirming that high-resolution features are the primary driver for instrument localization and pose estimation in narrow anatomical

corridors. In SurgPose, the improvements are more subtle but consistent as the YOLOv8 backbone is already strong at 92.2% / 95.3%, since SurgPose involves robotic instruments in a more uniform field of view. Finally, comparing HRNetV2p-pose to MID-POSE shows that adding the MIC head provides a slight refinement, by suppressing false positive detections. This demonstrates that while the encoder provides the spatial foundation, the dual-head design ensures semantic consistency.

Table 7: Class-agnostic performance comparison across backbones.

| Architecture | PitSurg - $mAP_{50-95}$ (Det/Pose) | SurgPose - $mAP_{50-95}$ (Det/Pose) |
|---|---|---|
| YOLOv8x-pose | 46.3% / 67.5% | 92.2% / 95.3% |
| HRNetV2p-pose | 84.6% / 84.5% | 93.5% / 96.5% |
| MID-POSE | 85.7% / 85.0% | 94.3% / 96.8% |

