# OpenReview forum: "MID-POSE: Multi-Instrument Detection and Pose Estimation in Endoscopic Surgery"
_MIDL.io/2026/Conference — MIDL 2026 Poster_

### Official Review · Reviewer_JR91 · 2026-01-06

**Confidence:** 4
**Preliminary Rating:** 4

**Summary:**

Authors propose a DL method for detecting multiple instruments while
also performing pose estimation from monocular endoscopic
videos. Furthermore, they also provide a dataset of pituitary surgery
videos including 2D keypoint annotations for various instruments. The
method also classifies different instruments, based on detection
outputs. First instruments, ignoring the type, are detected along
their keypoints using a YOLO-based approach. Then the detected ROIs
are classified to identify the type of instrument detected by the
YOLO. Notably, the main difference of the proposed method to YOLOv8 is
the encoder, for which authors use another existing architecture,
HRNETv2p. The proposed method is evaluated on both the newly
introduced dataset and an existing benchmark, i.e., SurgPose.

**Strengths:**

- The problem being solved here is quite relevant to MIS applications.
- The PitSurg dataset seems to be a great contribution to the field.
- The proposed architecture, possibly due to using higher-resolution
  feature maps, is able to classify different instruments much better.

**Weaknesses:**

- A comprehensive evaluation of a proposed model cannot be considered
  as a contribution. It is something we should all do whenever a new
  method is introduced. I suggest authors to remove the last point
  from the list of contributions.
- The technical novelty of the proposed method remains a bit low. To
  the best of my understanding, authors simply use YOLOv8-Pose
  approach and add an ROI-wise classification module on top of it.
- The total loss is composed of 4 different components each with a
  dedicated weight. Can authors discuss the sensitivity of the method
  and the training procedure to the choices of these weights.
- Use of different weight parameters on the different datasets reduces
  the enthusiasm for the overall method a bit. Can authors provide
  results keeping the same weights on both dataset as well? This can
  help readers to understand the sensitivity of the model training.

**Detailed Comments:**

Please see the list of weaknesses.

**Justification Of The Preliminary Rating:**

While the technical novelty may not be very high, authors present a useful algorithm that can help extract
richer information about the surgical instruments from the videos. The evaluation of the method is done quite
well in my opinion. Furthermore, authors present a new dataset with annotated key points for different
instrument types, which will be a great asset to the community.

**Questions To Address In The Rebuttal:**

+ Providing quantitative evaluation using the same weights ($\lambda$) on both datasets would help me judge the work better.
+ Providing class-agnostic detection accuracy for all the methods would be helpful to understand the contribution of the higher resolution feature maps.

---

> ### Author Response · Authors · 2026-01-25
> **Responds to Reviewer Reviewer JR91**
>
> We thank the reviewer for the constructive review. Please find below our responses on contribution, technical novelty, training sensitivity, and class-agnostic performance.
>
> 1. **Comprehensive evaluation is not a standalone contribution**
>
> We agree that evaluation is a standard requirement. Our intent was to highlight the creation of a benchmark across manual and robotic endoscopic MIS. We have revised the third contribution to clarify that it provides a benchmark for the community.
>
> 2. **Technical novelty**
>
> We thank the reviewer for this comment. MID-POSE is not a naive RoI classifier added to YOLOv8x-pose. We identify two bottlenecks in YOLO-style pose estimation: (1) progressive encoder downsampling discards fine spatial detail needed for precise box/keypoint localization; and (2) instrument classification is predicted densely per grid cell from the pyramid, making it vulnerable to local ambiguity and distractors and lacking explicit instance-level reasoning. We address (1) by using HRNetV2p, which preserves a high-resolution stream and enriches it via multi-scale fusion, improving spatial precision for bounding boxes and keypoints. We address (2) by introducing the MIC head, which performs instance-wise classification by fusing multi-level RoI features within each predicted box, followed by convolutional refinement and global pooling, improving semantic disambiguation and reducing false positives. The complementary roles of these components are supported by **Figs. 3–4**, the instrumentness-threshold sensitivity (**Table 3**), and stratified failure analysis (**Tables 5–6, Appendix B**), demonstrating consistent robustness gains across challenging visual conditions on both datasets.
>
> 3. **Sensitivity of loss weight choice**
>
> We thank the reviewer for raising this question. In pg.8 of the revised manuscript, we clarify that we kept the Ultralytics YOLOv8x-pose default weights fixed for the localization and pose terms, namely $\lambda_{\text{box}} = 7.5$, $\lambda_{\text{kpt}} = 12.0$, and $\lambda_{\text{inst}} = 1.0$. For dual-head models, the only additional weight is $\lambda_{\text{mic}}$, which controls the relative strength of the MIC loss.
>
> We evaluated sensitivity by sweeping $\lambda_{\text{mic}} \in \{10.0, 12.5, 15.0, 17.0\}$ on both PitSurg and SurgPose in **Table 2**. Performance varies smoothly and remains stable across the sweep: for both YOLOv8x-pose+MIC and MID-POSE, detection/pose mAP50-95 changes by at most ${\sim} 1-2$ points. On PitSurg, the best region is around $\lambda_{\text{mic}} {\sim} 10-12.5$, while on SurgPose, performance increases monotonically with larger $\lambda_{\text{mic}}$, consistent with SurgPose being more dominated by fine-grained semantic ambiguity.
>
> Overall, these results indicate that MID-POSE is not sensitive to a narrowly tuned $\lambda_{\text{mic}}$ and operates reliably over a broad range. For single-head baselines on SurgPose, we only adjusted $\lambda_{\text{cls}}$ (set to 40.0) to obtain meaningful multi-class separation under strong inter-class visual similarity.
>
> 4. **Provide results keeping the same weights on both dataset**
>
> We appreciate the reviewer’s suggestion. We therefore report results using one identical weight configuration on both PitSurg and SurgPose: $\lambda_{\text{box}} = 7.5$, $\lambda_{\text{kpt}} = 12.0$, $\lambda_{\text{inst}} = 1.0$, with a single shared $\lambda_{\text{mic}} = 15.0$ (**Table 2**, $\lambda_{\text{mic}} = 15.0$). Under this shared setting, MID-POSE gains over YOLOv8x-pose+MIC are essentially unchanged vs. Table 1: mAP50–95 Det/Pose +17.8/+13.2 (PitSurg) and +3.0/+3.5 (SurgPose), vs. +17.9/+13.6 and +3.0/+3.2, showing the gains are not driven by dataset-specific tuning.
>
> 5. **Class-agnostic performance to understand the contribution of the higher resolution feature maps**
>
> We thank the reviewer for the suggestion. Providing class-agnostic performance results effectively isolates the contribution of high-resolution feature maps from the classification logic. We have included these results in **Table 7** in **Appendix C**. In the PitSurg dataset, replacing the YOLOv8x-pose backbone with HRNetV2p results in a substantial gain in detection/pose mAP from 46.3% / 67.5% to 84.6% / 84.5%, confirming that high-resolution features are the primary driver for instrument localization and pose estimation in narrow anatomical corridors. In SurgPose, the improvements are more subtle but consistent as the YOLOv8x-pose backbone is already strong at 92.2% / 95.3%, since SurgPose involves robotic instruments in a more uniform field of view. Finally, comparing HRNetV2p-pose to MID-POSE shows that adding the MIC head provides a slightly refinement, by suppressing FP detections. This demonstrates that while the higher resolution encoder provides the spatial foundation, the dual-head design ensures semantic consistency.

---

### Official Review · Reviewer_HmRY · 2026-01-07

**Confidence:** 4
**Preliminary Rating:** 4
**Final Rating:** 4

**Summary:**

This paper introduces a new endoscopic surgery dataset for procedures at the skull base and a corresponding model architecture for detecting instruments and estimating their pose in such videos. The dataset of 26 videos extends publicly available datasets  by another type of procedure and a different location, and it has bounding boxes and keypoints for seven types of instruments annotated. The proposed architecture builds upon YOLOv8x-pose and makes some incremental changes: 1) a high-resolution feature encoder (motivated by this particular dataset's precision requirements) and 2) a multi-instrument classification head. These changes are evaluated on both the above new dataset as well as on the previously introduced SurgPose dataset, providing evidence that the former helps increasing spatial precision and the latter improves classification performance, thereby also reducing typical problems with false positives and class confusion.

**Strengths:**

1) The paper describes a solid work on surgical instrument detection and pose estimation on this particular procedure, with convincing results.
2) The changes made to the previously published architecture are well motivated and properly evaluated via a few ablation studies.
3) The plausible conclusion is that the observations can be translated to other tasks, in particular w.r.t. MIC head's improved classification.

**Weaknesses:**

1) The paper strongly builds upon previous work and does not reproduce all formulas or even explains all terms used, which reduces readability.
2) The weighting parameters of the loss function are empirically chosen and appear relatively arbitrary. In the worst case, the method could be sensitive to the exact choice of these weights (currently set to odd values such as 7.5, 12, 1, 10, 40, and 17, and varying for the two datasets).
3) The evaluation does not seem to test for statistical significance of the differences (Table 1 has just the highest numbers in bold, apparently).

**Detailed Comments:**

"a MIC head" is not properly introduced in the introduction.
"OKS" is not introduced at all, not even its meaning (abbreviation source)
p_alpha is not explained (formula (1))

Maybe help the international audience by adding some location information to "National Hospital for Neurology and Neurosurgery" (at least one of London, UK, UCL or similar) because it sounds more generic than it is if you're not familiar with it.

It is quite unusual to have weighting factors like 7.5, 12, or 17 in the loss function. Usually, the sum of the lambdas would be 1.0 or close to that. It seems as if you could scale the factors down and instead increase the learning rate accordingly.

The Figures are *much* too small.  Maybe the confusion matrices and IoU / OKS can be overlayed on the images if you want to make them bigger without requiring too much space.  Or you introduce an appendix.

If you need more space, maybe you can remove some numbers from the Quantitative results section, since it is not always clear what is the meaning of the numbers without looking at the table anyhow (e.g., after "modest gains over the YOLOv8x baseline"), and all numbers can already be found there.

Another way to save space could be to extract the label counts in the (split) PitSurg dataset into a table, possibly combined with the legend of Figure 2, so you don't have to repeat the instrument names so often.

I suggest to rephrase "RoI-pooled features", since you're using ROIAlign, not ROIPooling. (I understand that you apply global average pooling on the result.)  BTW: Did you also try global max pooling?  I could imagine that that even slightly improves the MIC performance.

The improved classification performance is particularly intriguing to me, yet it also makes the model larger and introduces new parameters.  Is that also the reason for the reduced batch size?  Wouldn't it be more fair to compare the classification performance of "equally complex" architectures?

**Justification Of Final Rating:**

The rebuttal has improved the manuscript, so it is somewhere between weak and strong accept.  Given the competition, I would not give it a perfect score yet, but I think the paper got good marks from the other reviewers as well, so it got a fair and positive rating.

**Justification Of The Preliminary Rating:**

I think the work is a nice contribution but the manuscript could be more legible. The contributions are mostly convincing, but there are some remaining doubts with respect to their generalisability (significance of improvements, model complexity, sensitivity of loss weighting).

**Questions To Address In The Rebuttal:**

See weaknesses, in particular statistical significance tests and more background on the loss weights could make the paper even more convincing.

Maybe also comment on the MIC head's size and model complexity (last detailed comment).

---

> ### Author Response · Authors · 2026-01-25
> **Responds to Reviewer HmRY**
>
> We thank the reviewer for the positive overall assessment and for the constructive comments on readability, loss weighting, statistical significance, and model complexity. Below, we address each point and describe the corresponding changes in the revised manuscript.
>
> 1. **Novelty concerns and clarity of definitions**
>
> We thank the reviewer for the remark. While MID-POSE can be seen as an incremental architectural improvement over the YOLOv8x-pose model, we argue that the proposed dual-head approach has a definite advantage on classification performance over the baseline, especially on robotic datasets in which the instruments are visually similar, as demonstrated by the results on SurgPose. We have added more details in the Loss Functions section (2.2) of the manuscript, expanding the definition of the Object Keypoint Similarity (OKS) metric, and clarified the weighting parameters used in the loss function at the end of the training protocol subsection.
>
> 2. **Location information for the National Hospital for Neurology and Neurosurgery**
>
> We thank the reviewer for pointing out that the original phrasing might be ambiguous for an international audience. We have updated the dataset description to read: “National Hospital for Neurology and Neurosurgery, London, UK”, and made explicit the connection to UCL in the author affiliations.
>
> 3. **Loss function weighting factors**
>
> We appreciate the comment and have clarified the loss-weight choices in the revised text. We explicitly state that most of the loss weights follow the Ultralytics YOLOv8x-pose defaults and were left unchanged: we kept $\lambda_{\text{box}} = 7.5$,
> $\lambda_{\text{kpt}} = 12.0$, and $\lambda_{\text{inst}} = 1.0$ as in the original implementation. The only weights we modified are:
>
> - $\lambda_{\text{cls}}$ for single-head models on the SurgPose dataset, which was empirically set at $40.0$ to obtain meaningful classification performance given the strong visual similarity between instrument types.
>
> - $\lambda_{\text{mic}}$ for dual-head models, which controls the relative strength of the MIC classification loss.
>
> For $\lambda_{\text{mic}}$, we now report in **Table 1** a small sensitivity study where we train dual-head models with several values and summarise the resulting detection and pose performance on both PitSurg and SurgPose. Empirically, lower values (e.g. 10.0) result in higher performance on PitSurg, while higher values (e.g. 17.0) favour SurgPose. Importantly, across this range the relative ordering of YOLOv8x-pose, YOLOv8x-pose+MIC, HRNetV2p-pose, and MID-POSE remains stable, indicating that our qualitative conclusions are not sensitive to a narrow choice of loss weights. We also explicitly note in the text that only the ratios between loss weights are meaningful and that globally scaling them is effectively equivalent to adjusting the learning rate for those terms.
>
> 4. **Test for statistical significance of the differences**
>
> We agree that statistical significance strengthens the empirical claims. Unfortunately, we did not have time to include this information in the short rebuttal period, but will investigate statistical significance in future work.
>
> 5. **Small figures**
>
> We thank the reviewer for the suggestion and have enlarged the figures for better readability in the revised manuscript.
>
> 6. **Rephrasing "RoI-pooled features", and use of global max pooling**
>
> We thank the reviewer for the suggestion and will investigate using global max pooling in the MIC head in future work.
>
> 7. **Concerns on classification performance, reduced batch size for models using the HRNetV2p backbone and fairness of comparison related to model complexity**
>
> We thank the reviewer for this important question. We trained the models using the HRNetV2p backbone with a reduced batch size to accommodate higher memory usage due to increased resolution relative to the baseline YOLOv8x-pose backbone. Regarding classification performance in MID-POSE, the baseline YOLOv8x-pose classification head has 7.38M parameters, while the MIC head only has 2.18M parameters. While we understand the reviewer's comment on the “increased conceptual complexity” of the architecture, this does not translate into a higher number of parameters. The primary factor for better classification performance of the MIC head is driven by the Object-Centric Features Extraction (computing feature representations on the object ROI), as opposed to the prediction grid, per-anchor approach of the baseline model.

---

### Official Review · Reviewer_yxTK · 2026-01-10

**Confidence:** 5
**Preliminary Rating:** 5

**Summary:**

This paper presents MID-POSE, a dual-head architecture designed for joint instrument detection and 2D keypoint pose estimation in the challenging context of endoscopic skull-base surgery. Recognizing that conventional mappers often struggle with the severe occlusion, truncation, and visual similarity of tools in pituitary procedures, the authors propose a decoupled approach: a class-agnostic dense head handles spatial localization (bounding boxes and keypoints), while a Multi-level Instrument Classification (MIC) head resolves semantic categories using RoI-pooled features from a high-resolution HRNetV2p encoder. A notable technical contribution is their extended keypoint visibility scheme, which allows the model to leverage partially annotated data and accommodate class-specific keypoint layouts.
To validate the model, the authors introduce the PitSurg dataset, comprising intraoperative frames from 26 clinical procedures featuring seven instrument types annotated with detailed keypoints. Experimental results on both PitSurg and the robotic SurgPose benchmark demonstrate that MID-POSE significantly outperforms state-of-the-art single-head baselines.

**Strengths:**

1. **Principled Architecture for Challenging Environments:** The paper addresses the high-complexity task of multi-instrument perception in monocular endoscopy, characterized by severe occlusion, truncation, and visual similarity. It used a well-justified "disentangled" design. By decoupling a class-agnostic dense head for localization from a dedicated RoI-based MIC classifier, the authors directly mitigate the performance trade-offs common in single-stage architectures when balancing spatial precision against fine-grained categorization.

2. **Robust Technical Implementation:** Several engineering choices make this disentangled approach effective in practice, notably the extended keypoint visibility scheme that accounts for "unannotated" states. This allows the framework to utilize class-specific layouts and partially labeled data without contaminating the pose supervision. Additionally, the two-stage training protocol and online RoI sampling strategy demonstrate a rigorous approach to handling class imbalance and hard negatives.

3. **Substantial Dataset Contribution:** The release of PitSurg which provides detailed bounding box and keypoint annotations for seven instrument types across 26 clinical pituitary procedures is a high-value contribution. It fills a significant gap in the literature for narrow-corridor neurosurgical endoscopy, where traditional robotic or abdominal datasets are not representative.

4. **Comprehensive Cross-Domain Validation:** The methodology is supported by strong empirical results on both PitSurg and the robotic SurgPose benchmark. The authors provide a particularly insightful analysis of the performance drivers, showing that while the HRNetV2p encoder is critical for spatial precision in pituitary scenes, the MIC head is the primary driver for resolving semantic ambiguities between visually similar robotic tool tips. This dual-domain success suggests the architecture is generalizable and not overfit to a specific surgical niche.

**Weaknesses:**

1. **Over-reliance on "Clean" Qualitative Examples:** The qualitative results (Fig. 4) provided in the manuscript primarily highlight ideal frames where instruments are clearly delineated. While the authors explicitly acknowledge that the model remains sensitive to extreme viewpoints, heavy overlap, and low-contrast instances near the image border , these specific "worst-case" failure modes are not systematically visualized. Providing a dedicated failure analysis would be more informative for assessing the system's robustness in the actual "hard regimes" it targets.

2. **Absence of Stratified Performance Analysis:** Given that occlusion and visual similarity are the central challenges motivating this work, the evaluation would be significantly strengthened by reporting performance conditioned on difficulty levels. Specifically, the paper lacks results stratified by bounding box area (for far-away or small tools), truncation levels at the image border, or the fraction of visible keypoints. Without such bins, it remains unclear whether the reported  gains are uniform or primarily driven by less challenging, well-centered instances.


3. **Interpretability of AP Metrics and Threshold Sensitivity:** While reporting  is standard for detection and pose tasks, these aggregated scores can be difficult to translate into practical surgical reliability. The inclusion of explicit success-rate summaries would provide a clearer indication of the system's operational consistency. Furthermore, the pipeline utilizes an instrumentness threshold gate $\tau$ of  at inference, yet the sensitivity of the model to this parameter $\tau$ is not explored. A detailed report on the  trade-offs across varying  values would be essential for understanding the model’s deployment behavior in distractor-heavy environments.

**Detailed Comments:**

1. **Add a compact “dataset datasheet” table (beyond narrative description).** The dataset section would be clearer and more reusable if PitSurg were summarized with concrete acquisition/quality details rather than mainly narrative text. For example, a small table could include: camera/endoscope type (if disclosable), native resolution and compression, fps (or frame sampling strategy), typical motion patterns (camera shake, rapid pans), lighting conditions, and approximate prevalence of blur/specular highlights/occlusion. This would make it much easier to understand how PitSurg differs from common public robotic datasets (e.g., da Vinci-style benchmarks) and why the proposed benchmarks are uniquely challenging.

2. **Clarify annotation conventions and difficulty statistics.** Since the paper emphasizes occlusion/truncation and class imbalance, it would help to report simple dataset-level statistics such as: distribution of bbox area (proxy for far-away tools), percentage of instances touching image borders (truncation), and a visibility breakdown of keypoints (visible vs occluded vs unannotated). These numbers would contextualize the reported AP/OKS results and reinforce the dataset’s value.

3. **Make “failure modes” easier to audit in the qualitative section.** Consider adding one extra qualitative panel showing “worst-(k)” frames (lowest IoU/OKS or highest FN/FP) and labeling the reason for failure (occlusion, tiny size, confusion with tissue/polyp-like structures, specular artifacts). This is a small addition but substantially improves trust in robustness claims without requiring new models.

**Justification Of The Preliminary Rating:**

I vote strong accept because the paper addresses a practically important and genuinely challenging problem, multi-instrument detection and 2D pose in monocular endoscopy under heavy occlusion, truncation, and high inter-class similarity, with a clean, well-motivated architectural idea and strong supporting evidence. The proposed class-agnostic detection–pose + MIC classification decomposition is conceptually simple yet effective, and the implementation details (visibility handling for class-specific keypoints, two-stage training, RoI sampling) make the disentanglement work in realistic data conditions.  The release of PitSurg (26 clinical pituitary procedures with seven tools and keypoint annotations) is itself a significant contribution that can anchor future benchmarking and ablation-driven progress in a domain not well covered by common robotic datasets.  Empirically, the method shows consistent and substantial gains on both PitSurg and SurgPose over strong single-head baselines, and the paper provides a credible component-level explanation of why different modules matter across domains (spatial precision vs class disambiguation).  While I would welcome more explicit worst-case and occlusion-stratified analysis, these are incremental improvements; the core idea, dataset value, and experimental results together make this a clear acceptance.

**Questions To Address In The Rebuttal:**

1. **Can you show systematic worst-case evidence (not only “clean” qualitative wins)?**
  Please add a small “worst-(k)” qualitative panel (e.g., the (k) frames with lowest IoU/OKS or highest FN/FP) for both PitSurg and SurgPose, and briefly categorize the dominant failure causes (heavy occlusion/complete overlap, tiny/far-away tools, truncation at borders, specular artifacts, confusion with tissue/polyp-like structures). This would materially strengthen the robustness narrative.

2. **How does performance change under occlusion and small/far-away instances?**
  Could you report stratified metrics (Det/Pose AP or success rates) binned by **bbox area** (small vs medium vs large), **truncation** (touching border vs not), and an occlusion proxy (e.g., fraction of keypoints marked visible)? A table or short plot would clarify whether the gains are uniform or mainly driven by easier examples.

3. **Can you provide success-rate thresholds and threshold sensitivity (deployment-relevant reliability)?**
  In addition to AP50–95, please report simple success rates such as ($\Pr(\text{IoU}\ge 0.5$)) and ($\Pr(\text{OKS}\ge 0.5$)), and include a brief sensitivity analysis to the instrumentness gating threshold (e.g., ($\tau\in{0.1,0.3,0.5}$)) showing FP/FN trade-offs. This would make the method’s practical reliability clearer and could change how strongly I view the results.

---

> ### Author Response · Authors · 2026-01-25
> **Response to reviewer yxTK**
>
> We thank the reviewer for the detailed, positive assessment and helpful suggestions. Please find below our point-by-point response to the comments.
>
> 1. **Qualitative results Fig. 4 only showing clean examples**
> We thank the reviewer for raising this point. We have clarified in **Section 3** (pg.7) of the revised manuscript that instruments in SurgPose are fully visible in all frames, with no occlusion, blur, specular highlights, or end-effector border truncation. By contrast, PitSurg contains these factors.
>
> 2. **Dataset difficulty statistics**
> We thank the reviewer for this suggestion. **Appendix A (Fig. 5)** reports the prevalence of PitSurg visual conditions, including clean frames and degraded regimes (partial/heavy occlusion, border truncation, blur, specular highlights, and extreme viewpoints). We do not report “small instruments” as a separate condition because, in PitSurg, small apparent size is a consequence of heavy occlusion or border truncation.
>
> 3. **Absence of stratified performance analysis**
> We thank the reviewer for this suggestion and agree that stratified evaluation is critical. **Appendix B (Tables 5 and 6)** provides a condition-stratified failure analysis of YOLOv8x-pose and MID-POSE on PitSurg and SurgPose. We define a failure as a GT instance that is either unmatched or matched with OKS < 0.5. The failure rate is the percentage of instances that fail within each condition. On PitSurg (**Table 5**), MID-POSE eliminates failures in Blur, Specular Highlights, and Clean (0.00%). It also improves Partial Occlusion from 6.75% to 0.78%. YOLOv8x-pose fails most under Heavy Occlusion (40.91%), Border Truncation (22.50%), and Extreme Perspectives (19.10%). MID-POSE reduces these to 10.00%, 6.25%, and 13.48%. On SurgPose (**Table 6**), YOLOv8x-pose fails in 30.1%, 43.6%, and 20.7% of instances under Fine-grained Semantic Ambiguity, Extreme Perspectives, and Clean. MID-POSE reduces these to 9.4%, 10.7%, and 0.36%. Overall, this stratified analysis shows that MID-POSE improves robustness consistently across all evaluated conditions, including the hardest regimes, rather than only on easier instances.
>
> 4. **Failure analysis and worst-case evidence**
> We agree that systematically exposing failure modes is important for auditing robustness. We added **Figures 6 and 7** in **Appendix B** to highlight the visual conditions that remain challenging for MID-POSE. For PitSurg, we focus on Partial Occlusion, Heavy Occlusion, Border Truncation, and Extreme Perspectives. For SurgPose, we focus on Fine grained Semantic Ambiguity and Extreme Perspectives. Within each condition, we select the instrument instance with the lowest OKS and show the corresponding frame to illustrate the dominant failure mode. Many of these failures arise from rare patterns that are absent or extremely underrepresented in the training data.
>
> 5. **Interpretability of AP metrics and threshold sensitivity**
> We appreciate the reviewer’s feedback on deployment-relevant interpretability. We added **Table 3** to the manuscript (pg.11), which reports the False Positives (FP) and False Negatives (FN) trade-off under different instrumentness thresholds $\tau$, together with success rates defined as $\Pr(\mathrm{IoU} \ge 0.5)$ for detection and $\Pr(\mathrm{OKS} \ge 0.5)$ for pose. MID-POSE demonstrates reduced threshold sensitivity over YOLOv8x-pose. Varying $\tau$ from $0.1$ to $0.5$ in MID-POSE maintains success rates above $90\%$ on PitSurg and above $79\%$ on SurgPose, while YOLOv8x-pose is much more sensitive, with success rates decreasing from above $90\%$ to around $60\%$. This suggests MID-POSE delivers more consistent performance over threshold shifts for deployment in surgical scenes.
>
> 6. **PitSurg acquisition details**
> We thank the reviewer for this suggestion. Acquisition details on the PitSurg dataset are presented in **Table 4**, and reported in the main content as "Frames are captured using a Hopkins Telescope with an AIDA storage system (Karl Storz Endoscopy, UK). The original videos were recorded at 720p, 24 FPS, and frames were sampled at 1 FPS for annotation".

---

### Author Rebuttal · Authors · 2026-01-25

**Rebuttal:**

We sincerely thank the reviewers and appreciate their valuable feedback. We have addressed each reviewer's comments separately in the Official Comments and provided a revised manuscript, with changes highlighted in orange. Please find below a summary of our point-by-point responses.

- We added a systematic failure analysis in **Tables 5** and **6** in **Appendix B** (pg.16), reporting instance-level failure rates across visual conditions on PitSurg and SurgPose, and visualizing worst-case examples in **Figures 6** and **7** (pg.17-18), showing MID-POSE’s gains persist in challenging regimes rather than being driven by easier instances.
- We performed a confidence-threshold sensitivity analysis in **Table 3** (pg.11). We swept the instrumentness gating threshold $\tau$ and reported FP/FN trade-offs together with interpretable success rates $\Pr(\mathrm{IoU} \ge 0.5)$ and $\Pr(\mathrm{OKS} \ge 0.5)$. The results show that MID-POSE is more stable than the baseline, consistently producing fewer false positives and higher success rates.
- We addressed the request for PitSurg dataset transparency by reporting difficulty statistics and providing available acquisition details in **Appendix A** (pg.15).
- We expanded section 2.2 (pg.5) to define OKS and more clearly describe the loss terms and training procedure.
- In the revised manuscript, we clarified that most loss weights follow the Ultralytics YOLOv8x pose defaults, and that dual head models introduce only one additional weight, $\lambda_{\text{mic}}$. We further included a $\lambda_{\text{mic}}$ sensitivity study in **Table 2** (pg.9), which indicates that the main conclusions remain stable across a range of values and are not dependent on narrow tuning.
- We enlarged and improved the readability of the figures in the revised manuscript.
- We clarified the fairness of comparisons, specifically on batch size and the classification heads' number of parameters.
- We clarified the third contribution and now present is as a benchmark for instrument pose estimation across manual and robotic endoscopy.
- We clarified that MID-POSE is a problem-driven disentangled design, highlighting the complementary roles of HRNetV2p for spatial precision and the MIC head for semantic consistency.
- We added class-agnostic performance results to isolate the impact of high-resolution features from classification logic in **Table 7** in **Appendix C** (pg.18).

Kind regards,

The authors

**Supporting Material:**

/attachment/0a0832686934892c2aa93320d27f63ce6bf40041.pdf

---

> ### Author Response · Authors · 2026-01-25
> **Rebuttal precisions**
>
> Please find below a more detailed description of the rebuttal.
>
> **Reviewer yxTK:**
> 1. We added a structured failure analysis in **Tables 5** and **6** in **Appendix B** covering PitSurg conditions including partial occlusion, heavy occlusion, border truncation, extreme perspectives, blur, specular highlights, and clean frames, and SurgPose conditions including semantic ambiguity, extreme perspectives, and clean frames.
> 2. We quantified robustness under these challenges by reporting failure rate in **Tables 5** and **6** for each challenge category, demonstrating that MID-POSE improves performance consistently on hard cases rather than only on easier instances.
> 3. We performed a confidence-threshold sensitivity analysis in **Table 3**. We swept the instrumentness gating threshold $\tau$ and reported FP/FN trade-offs together with interpretable success rates $\Pr(\mathrm{IoU} \ge 0.5)$ and $\Pr(\mathrm{OKS} \ge 0.5)$. The results show that MID POSE is more stable under changes in $\tau$, and it consistently produces fewer false positives and higher success rates than the baseline.
> 4. We addressed the request for PitSurg dataset transparency in our response by reporting difficulty statistics, including counts per challenge category, and by providing the available acquisition details in **Appendix A**.
>
> **Reviewer HmRY**:
> 1. We expanded the Loss Functions/training protocol description (Sec. 2.2) to define OKS, clarify notation, and more clearly describe the loss terms and training procedure.
> 2. In the revised manuscript, we clarified that the loss weights for localization and pose follow the Ultralytics YOLOv8x pose defaults, and that dual head models introduce only one additional weight, $\lambda_{\text{mic}}$. We further included a $\lambda_{\text{mic}}$ sensitivity study in **Table 2**, which indicates that the main conclusions remain stable across a range of values and are not dependent on narrow tuning.
> 3. We enlarged and improved the readability of the figures in the revised manuscript.
> 4. We clarified the fairness of comparisons: aside from using a smaller batch size for HRNet-based models due to higher activation memory from high-resolution feature maps, all other training and evaluation settings are identical.
>
> **Reviewer JR91**:
> 1. We revised the third contribution to emphasise the benchmark across manual and robotic endoscopy, rather than presenting the evaluation as a standalone contribution.
> 2. We clarified that MID-POSE is a problem-driven disentangled design, with a class-agnostic dense head that focuses on geometry, and a multi-level RoIAlign MIC head with hard negative sampling for semantic disambiguation. We also highlighted the complementary roles of HRNetV2p for spatial precision and MIC for semantic consistency, supported by the analyses in **Tables 2** and **7**.
> 3. We added a $\lambda_{\text{mic}}$ sweep in **Table 2** and reported results using a shared set of loss weights across PitSurg and SurgPose for dual head models, demonstrating that our conclusions do not rely on dataset-specific re-weighting of the loss terms.
> 4. We added class-agnostic performance to isolate the impact of high-resolution features from classification logic in **Table 7** in **Appendix C**.

---

### Meta-Review · Area_Chair_ecFF · 2026-02-08

**Recommendation:** Accept (Oral)
**Confidence:** 5

**Metareview:**

This is an interesting work which proposed a multi-branching architecture designed for joint instrument detection and 2D keypoint pose estimation in the challenging context of endoscopic skull-base surgery. The main contribution of this work is that the proposed method can effectively parse multiple instruments in the same scene even with monocular and multiple tasks. The experiment results are also convincing. My recommendation of this work is accept.

---

### Decision · Program_Chairs · 2026-02-13

Accept (Poster)